# Iterative Monte Carlo Tree Search
# for Neural Architecture Search

**Mehraveh Javan Roshtkhari**[1]  **Matthew Toews**[1]  **Marco Pedersoli**[1]

[1]École de technologie supérieure (ÉTS), Montréal, Canada

**Abstract**   Recent work has shown Monte-Carlo Tree Search (MCTS) as an effective approach for Neural Architecture Search (NAS) in producing competitive architectures. However, the performance of the tree search is highly sensitive to the node visiting order. If the initial nodes are highly discriminative, good configurations can be efficiently found with minimal sampling. In contrast, non-discriminative initial nodes require exploring an exponential number of nodes before finding good solutions. In this paper, we present an iterative NAS approach to jointly train the recognition model with MCTS and learn the optimal node ordering of the tree. With our approach, the order of node visits in the tree is iteratively refined based on the estimated performance of the nodes on the validation set. With this approach, good architectures are more likely to naturally emerge at the beginning of the tree, improving the search process. Experiments on two classification benchmarks and a segmentation task show that the proposed method can improve the performance of MCTS, compared to state-of-the-art MCTS approaches for NAS.

## 1 Introduction

Monte-Carlo Tree Search (MCTS) is a powerful approach for non-differentiable problems, particularly those involving discrete actions (Browne et al., 2012; Costa and Pedreira, 2023). However, sampling efficiency is crucial for MCTS to minimize unnecessary exploration and achieve faster convergence to good solutions (Świechowski et al., 2023). Poor sampling efficiency, especially with large search spaces and hard-to-distinguish branches, can hinder its efficient application. MCTS is a compelling approach for Neural Architecture Search (NAS) due to its inherent exploration-exploitation mechanism and ability to handle imperfect evaluations. This is particularly important in one-shot NAS methods, where the recognition model training and architecture search are performed simultaneously.

One-shot methods based on weight-sharing (Pham et al., 2018), reduce the cost of evaluating candidate architectures. Essentially, an over-parameterized model containing all possible architectures, called "supernet", is trained. The supernet is then used to estimate the performance of an architecture by inheriting the weights. This eliminates the need to train individual architectures. However, shared weights can introduce bias and interference during supernet training and lead to rank inconsistency for the sub-nets compared to standalone training performance. (Yu et al., 2019; Bender et al., 2018; Zhao et al., 2021a). Jointly performing MCTS and supernet training may benefit the search, by gradually reducing the number of sampled architectures with shared weights.

Several works have explored MCTS for NAS, with various designs and search methods (Wang et al., 2021; Zhao et al., 2021b, 2024; Su et al., 2021; Roshtkhari et al., 2025). Some have used MCTS only for the search phase (Wang et al., 2021; Zhao et al., 2021b), while others have utilized it for both training and search (Su et al., 2021; Roshtkhari et al., 2025) as in our case.

Design of the search tree is crucial for NAS efficiency: Su et al. (2021) use a manual tree design, considering each layer of the CNN as a level in the tree, with operation choices as branches, while Roshtkhari et al. (2025) propose to learn the tree structure in unsupervised manner during

supernet training. Wang et al. (2021) and Zhao et al. (2021b) partition the search space based on the performance of a trained supernet.

In general, applying MCTS for NAS without additional constraints and regularization leads to poor performance (Su et al., 2021). The manual tree design proposed by Su et al. (2021) requires additional regularization to compensate for low sampling rates of nodes. This regularization (soft independence assumption) undermines the joint contribution of operations in layers by sharing reward information among nodes (Roshtkhari et al., 2025).

A promising solution is to improve branching quality by learning an optimized tree structure from the data (Roshtkhari et al., 2025). As it is computationally prohibitive to sample the entire tree with high frequency, an optimized tree should prioritize sampling regions with high ground truth performance more frequently.

A reasonable approach is partitioning "good" and "bad" regions of the search space by clustering based on estimated performance. However, the quality of this pre-ordered tree relies on accurate rankings of architectures, which depend on the quality of the sampling, generating a typical chicken-and-egg problem. Previous approaches (Wang et al., 2021; Zhao et al., 2021b) tackle this problem by learning how to separate the search space while performing the architecture search. In those works, the recognition model is given, assuming it already provides good estimations, which renders the factorization of the search space based on static estimations. Nevertheless, when the recognition model is trained while learning the search space hierarchy, the problem becomes much more complex and does not allow for static solutions.

In this work, we propose a simple approach in which the tree structure is reorganized as the supernet training progresses. At each iteration, MCTS is used to guide supernet training, and the performance estimated from this supernet is used to refine and reorganize the search tree. We show that, while initial performance estimates may not be a reliable measure for constructing the search tree, an iterative application of MCTS and tree reorganization can gradually guide the search towards high-performing architectures by gradually increasing their sampling rates.

The main contributions of our work are as follows:

- We present a new method of partitioning NAS search space into a search tree, based on performance estimates obtained from the supernet. We show that by iterative application of MCTS and tree reorganization, we can obtain competitive architectures without the prohibitive cost or constraints of previous methods.

- We show that with a careful balance of exploration and exploitation, the number of iterations needed is small and the overhead cost is negligible.

- We empirically validate our method for two computer vision tasks of image classification and semantic segmentation and on three datasets. We show that compared to other MCTS-NAS methods that perform supernet training and MCTS jointly, our approach achieves competitive performance without regularization and with linear computational complexity.

## 2 Related Work

### 2.1 Monte-Carlo Tree Search for NAS

AlphaX (Wang et al., 2019) was a significant early work that utilized Upper Confidence applied to Trees (UCT) (Auer et al., 2002) for MCTS-NAS. They proposed the use of MCTS to balance exploration and exploitation and increase the sample efficiency for NAS. They train a predictive model (Meta-DNN) to estimate the accuracy of architectures based on their encoding and guide MCTS. However, training a high quality Meta-DNN, while reducing architecture evaluation cost, requires sufficient data (architecture-prediction pairs) which adds computational overhead. Among methods that factorize the search space manually, TNAS (Qian et al., 2022) proposed to improve

exploration by partitioning it into two tree structures (operation and architecture). They utilized a bi-level breadth-first search algorithm to navigate the search space more efficiently. However, the proposed operation tree is unbalanced (Le et al., 2024) and the breadth-first search process requires additional training epochs as the network deepens.

Su et al. (2021) apply MCTS on a macro search space, and construct the tree manually by considering each layer of the CNN as a level of the tree and branching on operations. To compensate for low sampling rates of tree leaves, they propose a regularization method (node communication). However, this regularization assumes soft node independence (Roshtkhari et al., 2025), which further couples operations that share weights by additionally sharing reward information among them. Other works aim to learn the tree structure from data. Wang et al. (2021) uses performance of architectures, with weights inherited from a pre-trained supernet, to partition search space into "good" and "bad" regions. Zhao et al. (2021b) aimed to find architectures close to the Pareto frontier for multi-objective NAS. They use hyper-volume to iteratively partition the search space and MCTS to account for partitioning errors. However, both these methods decouple supernet training from MCTS by relying on a fixed pre-trained supernet or benchmarks. Roshtkhari et al. (2025) use supernet estimations to learn tree structure by hierarchal clustering of architectures using their output (functional) distances.

## 2.2 NAS for Semantic Segmentation

The main focus of NAS for computer vision has been on image classification task with CNNs, leaving other tasks (e.g. dense prediction) less developed (Mohan et al., 2023). NAS for semantic segmentation is more challenging: Compared to classification, it requires higher computational cost and memory since the input images and feature maps generally have higher resolution, and the architectures are deeper and more complex to enable per pixel prediction. In the case of medical images, data can be 3D (Ali et al., 2024), while some applications require real-time inference. Therefore, a good trade-off between performance and efficiency is essential.

Furthermore, fewer benchmarks (Duan et al., 2021; Mehta et al., 2022; Zhao et al., 2024) are available for segmentation task (Chitty-Venkata et al., 2023), adding to the computational cost of evaluating NAS methods and reproducibility. Contrary to classification, the architectures require a decoder or task-specific head, which can be searched separately (Chen et al., 2018a; Ghiasi et al., 2019; Xu et al., 2019) or jointly (Guo et al., 2020a; Yao et al., 2020) with the encoder part. To improve efficiency, many works use differentiable methods (Liu et al., 2019; Saikia et al., 2019; Xu et al., 2019; Guo et al., 2020a), while others use reinforcement learning or evolutionary algorithm (EA) (Ghiasi et al., 2019; Du et al., 2020; Wang et al., 2020b; Bender et al., 2020).

Auto-DeepLab (Liu et al., 2019), one of the most prominent NAS works for segmentation, proposed to use a bi-level (hierarchical) search space (macro-level: resolution and channels ; micro-level: cell or blocks) and applied DARTS iteratively to these two levels. Application of differentiable approach resulted in significant reduction in computational cost compared to DPC (Chen et al., 2018a), making NAS feasible for segmentation. DCNAS (Zhang et al., 2021) builds upon this, constructing a densely connected search space and using path and channel level sampling to reduce the computational cost. This enabled to directly search on the target task without using a proxy task or dataset.

Several works aim for real-time applications by applying latency constraints (Shaw et al., 2019; Chen et al., 2019; Lin et al., 2020). The latency of each architecture is often estimated and incorporated into the loss function for a differentiable search. Other works apply multi-objective NAS for efficient segmentation (Lu et al., 2022; Yu et al., 2024). Another approach is to extend the application of zero-cost proxies developed for classification (Abdelfattah et al., 2021; Lee and Ham, 2024) to segmentation. SasWOT (Zhu et al., 2024) proposes to use EA and learn a zero-cost proxy specifically for semantic segmentation. This zero-cost proxy is then used to evaluate architectures at initialization, greatly reducing the computational cost of NAS.

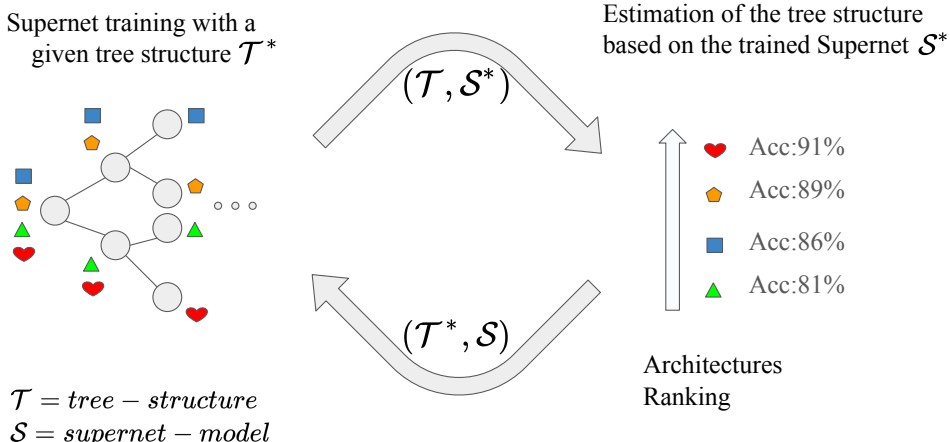

Supernet training with a
given tree structure $\mathcal{T}^*$

Estimation of the tree structure
based on the trained Supernet $\mathcal{S}^*$

$(\mathcal{T}, \mathcal{S}^*)$

$(\mathcal{T}^*, \mathcal{S})$

Acc:91%

Acc:89%

Acc:86%

Acc:81%

$\mathcal{T} = tree - structure$
$\mathcal{S} = supernet - model$

Architectures
Ranking

Figure 1: Overview of our iterative MCTS. At each iteration, the tree structure $\mathcal{T}^*$ is provided, and a new supernet $\mathcal{S}$ is trained. With the trained supernet $\mathcal{S}^*$, a new tree structure $\mathcal{T}$ is estimated based on the performance (accuracy) of architectures on validation data using supernet. These iterations are repeated until convergence.

## 3 Method

We propose an iterative MCTS algorithm where the tree structure is refined at each iteration using accuracy estimates of the leaf nodes on a validation set. This progressive tuning of the tree structure allows compensating for noisy and inaccurate estimation obtained from the supernet. Our iterative algorithm is depicted in Fig. 1.

During the recognition model training, MCTS samples from the supernet, while estimating the probabilities of each node, such that the architectures with higher estimated accuracy are generally sampled more often. With the constructed MCTS, we estimate the accuracy of each architecture to establish a comprehensive ranking. This ranking is then used to build a new search tree by simply organizing the architectures based on their validation scores. Given the new tree structure, we can repeat the first phase of training. This iterative procedure continues for a predefined number of iterations. In the following subsections, we will explain each part of this algorithm in detail.

### 3.1 Initialization

In order to start the iterative procedure, we need to provide an initial tree structure, $\mathcal{T}_{init}$, and an initial model for the supernet, $\mathcal{S}_{init}$. In our experiments, we initialize the supernet with single-path uniform training: for each minibatch, we train the supernet by randomly sampling an architecture $a$ from the search space. With $\mathcal{S}_{init}$, we can compute the accuracy of each architecture on the validation data. This allows us to rank the architectures and provides an initial structure for the tree $\mathcal{T}_{init}$.

### 3.2 Training the Supernet by Sampling with MCTS

We define a supernet $\mathcal{S}$ as a recognition model that includes all operations and parameters required to build any feasible architecture within our search space. We also use a tree structure $\mathcal{T}^*$ that defines how the set of architectures is divided into smaller subgroups, going from entire search space at root to leaf nodes representing individual architectures. Given a tree structure $\mathcal{T}^*$, each leaf

node corresponds to an architecture $a$. Architecture $a$ is sampled by traversing $\mathcal{T}^*$ from the root and making node selections among successors (children). A common method to balance exploration and exploitation for node selection is to use UCT (Kocsis and Szepesvári, 2006). For each visited node $i$, two values are recorded: visit count $n(i)$ and the average reward of the node $\tilde{A}(i)$. The UCT value is then calculated as:

$$UCT(i) = \tilde{A}(i) + C \sqrt{\frac{log(n_p(i))}{n(i)}} \tag{1}$$

and $n_p(i)$ is the function representing the number of times $i$'s parent node was visited, and $C \in \mathbb{R}_+$ is the constant that balances exploitation (the first term) and exploration (the second term). As the node with the highest UCT score is generally selected among sibling nodes, we encourage further exploration by applying Boltzmann sampling (Painter et al., 2024). Therefore, the probability of a sampling node $i$ is calculated as:

$$P(i) = \frac{exp(UCT(i)/T)}{\sum_j exp(UCT(j)/T)} \tag{2}$$

where the summation is performed over all sibling nodes of $i$. The temperature term $T$ controls the probability distribution, with $T \rightarrow 0$ corresponding to categorical distribution.

Using single-path approach (Guo et al., 2020b), the supernet $\mathcal{S}$ is trained on sampled architecture $a$ for one iteration. To avoid overfitting on training set, we use validation performance to calculate the reward for each architecture. At each iteration, $\mathcal{S}$ is trained on one minibatch of training data and then evaluated on one minibatch of validation data to yield performance $A(a)$. This value is then backpropagated to update the rewards along the selected path in the tree. The reward is calculated using a weighted moving average:

$$\tilde{A}(i) \leftarrow \beta \cdot \tilde{A}(i) + (1 - \beta) \cdot A(a) \tag{3}$$

where $\beta \in [0, 1]$ is the weighting factor. The process of sampling, training, evaluation, and reward backpropagation is repeated for a specified number of epochs. Since a static tree is used, the expansion and rollout phases of traditional MCTS are omitted. Once this phase is finished, the trained supernet $\mathcal{S}^*$ is passed to the next phase to update the tree structure. To select the final architectures for evaluation, we use Equation 1 with $C = 0$, since we do not need exploration in this stage.

### 3.3 Updating the Tree Structure

Our goal is to leverage the trained supernet $\mathcal{S}^*$ to construct an improved hierarchy that guides exploration towards promising nodes. By placing superior nodes in preferred paths, fewer nodes need to be explored, allowing the allocation of resources to these nodes and faster convergence of search. To achieve this, we construct a binary tree that represents this hierarchical structure based on the ranking of leaf nodes. Given a trained supernet $\mathcal{S}^*$, we rank sampled architectures based on their validation performance (accuracy). Since evaluating on the entire validation set is computationally expensive, we approximate the performance by evaluating architectures on only a few minibatches of validation data. With this ranking, a bottom-up approach then iteratively merges the two nodes (architectures or existing clusters) with the lowest average ranks, and the process is continued until a new tree, $\mathcal{T}^*$ is constructed.

### 3.4 Iterative MCTS

We treat the tree structure $\mathcal{T}$ as a heuristic, which provides a good starting point, but is updated and reorganized at each iteration of MCTS as new information comes in. At each iteration of

MCTS, with a good balance of exploration/exploitation, value estimates of the nodes are refined, reflecting the algorithm's learned understanding of the tree. Therefore, at each iteration, we update $\mathcal{T}$ based on the newly acquired ranking. In section 4.1, we use the same ranking criteria for tree initialization and analyze alternatives in section 4.1.1.

Algorithm 1 outlines our approach. To start iterative MCTS method, an initial tree structure and supernet are provided to the algorithm (Section 3.1). The main loop is composed of a first loop in which a branch of the tree is stochastically sampled based on node probabilities (Equation 2). This process selects an architecture $a$, which is then used for training the supernet $\mathcal{S}$ for one minibatch. The probabilities of the tree $P$ are then updated based on the accuracy of the given architecture on a validation minibatch. Finally, after several training iterations, the architectures are ranked and used to update the tree structure, and the training of the supernet is started again with the new tree structure.

In a static tree for MCTS, the UCT does allow for some exploration of initially misclassified "bad" branches of the tree (by tuning hyperparameter $C$ in Equation 1). However, the hierarchy does not have a chance to improve itself based on the information learned from this exploration; a potentially good architecture can get permanently placed in a bad region. Manual tree design (Su et al., 2021) or relying on potentially inaccurate supernet performance estimates to partition search space (Wang et al., 2020a), also do not guarantee an optimized tree structure. We propose that with well-balanced exploration and exploitation, good architectures can be identified and the tree can be restructured iteratively to prioritize these architectures. To achieve this, we propose to re-rank architectures and reorganize the search tree in each iteration of MCTS.

---

**Algorithm 1**: Simplified pseudo-code of our iterative MCTS.

---

$M$: number of MCTS iteration; $K$: iteration of each MCTS; $\mathcal{X}_t, \mathcal{X}_v$: minibatches of training
   and validation data; $\mathcal{S}_{init}$: Initial supernet, $\mathcal{T}_{init}$: Initial tree structure.
$m = 0, k = 0$
$\mathcal{T} \leftarrow \mathcal{T}_{init}, \mathcal{S} \leftarrow \mathcal{S}_{init}$
**while** $m \leq M$ **do**
    $P \leftarrow init(\mathcal{T})$  #initialize the tree probabilities with the new structure
    **while** $k \leq K$ **do**
        $i \leftarrow i_{root}$  #start from the root node
        $path = []$  #keep the entire path to backpropagate probabilities
        **while** $i\ not\ leaf$ **do**
            $path.add(i)$
            $a = sample(P(i))$  #sample based on the tree probabilities
            $update(n(a))$  #update count for child node
            $a \leftarrow i^*$
        **end**
        $\mathcal{S}.train(\mathcal{X}_t, a)$  #train the supernet
        $Acc(a) = \mathcal{S}.evaluate(\mathcal{X}_v, a)$  #estimate the accuracy of the architecture $a$
        $P \leftarrow backpropagate(Acc(a), path)$  #update the tree probabilities
    **end**
    $\mathcal{T} \leftarrow Rank(Acc)$  #rank the architectures based on accuracies and build the new tree
**end**
**Output**: Best architecture from $\mathcal{T}$ by sampling with $C = 0$

---

## 4 Experiments

In this section, we first apply our iterative MCTS for NAS on two image classification search spaces: Pooling search space (Roshtkhari et al., 2023) on CIFAR10 dataset and NAS-Bench-201 (Dong and

Table 1: Comparison results on CIFAR-10 using pooling search space (Roshtkhari et al., 2023).

| Method | Accuracy | | Search Time |
| --- | --- | --- | --- |
| | Best | Average | |
| Default (Resnet20) | 90.52 ± 0.15 | - | - |
| DARTS (Liu et al., 2018) | 89.23 ± 0.08 | - | 12 |
| DARTS + GAEA (Li et al., 2020) | 89.12 ± 0.10 | - | 12 |
| Balanced Mixtures (Roshtkhari et al., 2023) | 91.55 ± 0.12 | - | 6 |
| Uniform Sampling | 90.52 ± 0.15 | 90.40 ± 0.08 | 1.5 |
| Boltzmann Sampling | 90.88 ± 0.08 | 90.51 ± 0.12 | 3 |
| MCTS-default | 90.85 ± 0.12 | 90.57 ± 0.21 | 2 |
| MCTS-prioritized (Su et al., 2021) | 91.78 ± 0.11 | 91.42 ± 0.11 | 2 |
| Iterative MCTS (ours) | 91.83 ± 0.12 | 91.81 ± 0.02 | ∼ 2 |
| Best Architecture | 92.02 ± 0.18 | - | - |

Yang, 2020) on ImageNet-16-120 dataset, and perform ablation studies on these tasks. We then show a promising application of our method to a semantic segmentation task in a trellis search space inspired by Auto-DeepLab (Liu et al., 2019). In our experiments, to obtain a higher quality ranking, we evaluate architectures on few minibatches of validation data. In ablation studies, we show that this approach provides higher quality final architectures.

## 4.1 Image Classification

We performed experiments on two NAS benchmarks: the Pooling benchmark (Roshtkhari et al., 2023) and NAS-Bench-201 (Dong and Yang, 2020). We compare our methods with non-hierarchical and MCTS methods. Uniform sampling serves as a baseline for comparison. Boltzmann softmax sampling (Cesa-Bianchi et al., 2017) is a simple method that offers a biased search by adjusting uniform probability distribution, with a temperature hyperparameter controlling the balance of exploration/exploitation (See Appendix A.2).

For comparison with MCTS methods, we consider MCTS-default (the manual design proposed by Su et al. (2021)) and MCTS-prioritized (same manual design with their proposed additional regularization). Additionally, for both benchmarks, we compare with the differentiable method DARTS (Liu et al., 2018). For the Pooling benchmark, we also report results of DARTS+GAEA (Li et al., 2020) and "Balanced Mixtures" (Roshtkhari et al., 2023), a method that learns non-hierarchical search space partitioning with a specialized supernet per partition. For NAS-Bench-201, we compare with various methods: GDAS (differentiable), ENAS (RL), RSPS (random search), and NASWOT (zero-cost proxy).

The Pooling search space is a small yet challenging search space featuring Resnet-like (He et al., 2015) architectures, with the goal of optimizing feature map sizes at each layer. Due to notable low rank correlation between supernet estimates and ground truth, it is a suitable benchmark for demonstrating the effectiveness of our approach. As presented in table 1, our method outperforms its counterparts with similar or less search time. We also note that in this benchmark, several methods achieve performance close to the upper bound, and therefore, significant net improvements in accuracy is not possible. We report results on NAS-Bench-201 dataset for ImageNet-16-120 in table 2. In this benchmark, our method outperforms other common NAS methods.

### 4.1.1 Ablation Studies.

**Number of MCTS iterations**. We analyze the optimal number of iterations for our method on Pooling benchmark in figure 2 (left). For each additional iteration of MCTS, total training steps is

Table 2: Comparison results on ImageNet-16-120 using NAS-Bench-201 (Dong and Yang, 2020). Results for non-MCTS methods are taken from papers.

| Method | Accuracy | | Relative Search Time |
|---|---|---|---|
| | Best | Average | |
| DARTS (Liu et al., 2018) | - | 16.4 | 3 |
| ENAS (Pham et al., 2018) | - | 16.3 | 3.7 |
| RSPS (Li and Talwalkar, 2020) | - | 31.1 | 2.1 |
| GDAS (Dong and Yang, 2019) | - | 41.8 | 8 |
| NASWOT (Mellor et al., 2021) | - | 38.3 | - |
| Uniform Sampling | 31.2 | 31.0 ± 0.2 | 3.8 |
| Boltzmann Sampling | 31.1 | 30.8 ± 0.3 | 4.5 |
| MCTS-default | 41.7 | 40.2 ± 0.4 | 4.1 |
| MCTS-prioritized (Su et al., 2021) | 41.7 | 41.4 ± 0.2 | 3.1 |
| Iterative MCTS (ours) | 42.2 | 41.9 ± 0.2 | 3.1 |
| Best Architecture | 47.3 | - | - |

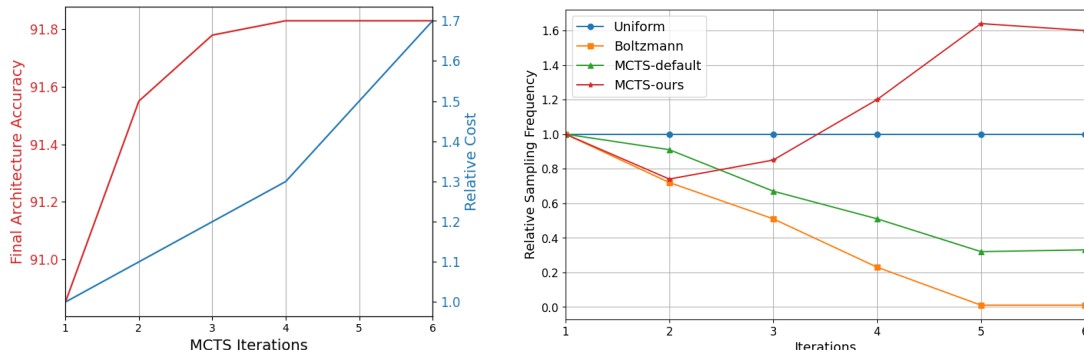

Figure 2: (left) Effect of the number of MCTS iterations on the quality of found architectures. Multiple iterations of MCTS can find superior architectures compared to a single iteration, with only a slight increase in training time. (right) Relative sampling frequency of the top-5 architectures. The x-axis corresponds to the number MCTS iterations used in our method. For other methods, we use equivalent time during training. The first iteration corresponds to the uniform sampling for warm-up or pretraining of various methods.

slightly increased to allow adequate sampling rate for nodes. Iterative MCTS is clearly superior to non-iterative MCTS in terms of the found architecture, and there seems to be an optimal number of iterations, beyond which the final result do not improve. The number of iterations and training cost can be treated as a trade-off when the training budget is limited.

**The effectiveness of iterative MCTS.** To demonstrate that the iterative process helps in guiding the search toward promising architectures, we calculated the sampling frequency of the top-5 architectures in the Pooling benchmark throughout the supernet training. In figure 2 (right), we compare sampling frequency of several methods. The frequency is recorded and averaged over 5 runs for each method. For fairness, we considered a fixed number of training iterations for all methods. For Boltzmann and MCTS-default, where the search converges to suboptimal configurations, the frequency unsurprisingly decreases. By increasingly sampling architectures outside top-5, supernet is guided towards those architectures, leading to lower final architecture

Table 3: (left) Comparison of ranking correlation between ground truth accuracy and supernet prediction for top-10 architectures. (right) Comparison of final accuracy based on evaluation on $B$ minibatches of validation data and using moving average of accuracy (equation 3).

| Method | Kendall's tau | Performance Metric | Final Accuracy |
|---|---|---|---|
| Uniform Sampling | 0.14 | Validation Accuracy ($B = 1$) | 41.6 |
| Boltzmann Sampling | 0.11 | Validation Accuracy ($B = 2$) | 42.1 |
| MCTS-default | 0.32 | Validation Accuracy ($B = 3$) | 42.2 |
| Iterative MCTS (ours) | 0.41 | Moving Avg. Accuracy | 40.5 |

Table 4: Comparison of searched architectures with various NAS methods for semantic segmentation task on Cityscapes dataset. Search space consists of macro (network) level of Auto-DeepLab (Liu et al., 2019). For consistency we report results from our own implementation.

| method | Best | Average | Time (GPU Days) |
|---|---|---|---|
| Uniform Sampling | 53.11 | 50.42 | $\sim 4$ |
| MCTS-prioritized (Su et al., 2021) | 75.32 | 73.1 | $\sim 3$ |
| Auto-DeepLab-S (Liu et al., 2019) | 76.91 | 76.73 | - |
| Iterative MCTS (ours) | **77.11** | **77.07** | $\sim 2.5$ |

accuracy. Our method show gradual increase in sampling these architectures, demonstrating its ability to improve sampling rate for good architectures.

**Rank-preserving ability.** While our method is able to concentrate training on promising architectures, we further analyze the ability of the supernet to distinguish and correctly rank these architectures correctly. In other words, we would like to know if the trained supernet has high enough quality to distinguish the top architectures. Calculating Kendall's tau coefficient of top architectures can indicate rank preservation (Zhang et al., 2024). In table 3 (left), we investigate the rank correlation of the top-10 architectures in Pooling benchmark with ground truth ranking by calculating Kendall's tau coefficient. We note that compared Boltzmann and MCTS-default our method achieves better rank correlation.

**Ranking metric for tree reconstruction.** At each iteration of MCTS, the performance of architectures need to be evaluated to calculate ranking. Evaluating on few minibatches of validation data provides a balance of accuracy and computational cost. Alternatively (at $M > 1$) one can use moving average from equation 3 which provides a smoother estimates and does not require further validation. In table 3 (right) we compare various metrics for NAS-Bench-201.

## 4.2 Semantic Segmentation

To evaluate our approach for segmentation task, we perform our experiments on a search space inspired by Auto-DeepLab (Liu et al., 2019). This search space is based on DeepLabV3+ (Chen et al., 2018b), in which the encoder consists of a found architecture and the decoder is not altered. Auto-DeepLab uses a bi-level search space (macro and micro) and gradient descent (DARTS) to optimize both levels iteratively. In this work, we focus on the network skeleton (macro) portion of the search space. This reduces the search space size from $10^{19}$ to $2.9 \times 10^{4}$. For semantic segmentation, mean Intersection Over Union (mIOU) is the standard performance metric; therefore we replace accuracy $A(a)$ in equation 3 with mIOU for this task. We report results of our implementations of several methods in table 4 on Cityscapes (Cordts et al., 2016) dataset.

## 5 Conclusion

In this paper, we present a novel MCTS approach for NAS. We developed an iterative method that progressively refines the search space hierarchy based on the observations from the supernet. Compared to previous applications of MCTS for NAS, the proposed approach does not use any specific knowledge to refine the search, making it more general and flexible. Our proposed approach iteratively updates the structure of the tree to favor high-accuracy architectures. We empirically evaluated our method on two classification tasks (CIFAR-10 on Pooling benchmark, ImageNet-16-120 on NAS-Bench-201) and a semantic segmentation on Cityscapes dataset.

**Limitations**. The proposed approach shows how to improve the performance of a supernet by iteratively estimating the best sampling tree and the recognition model. However, the approach assumes that the iterative refinement starts from a relatively good initialization of the supernet. In our experiments, we use as initialization a supernet trained with uniform sampling which performed adequately well. Nevertheless, if the initial recognition model ranking estimates are not sufficiently correlated with the true architecture ranking, the self-refining approach may not lead to improved results.

**Acknowledgements**. This work was supported by the Natural Sciences and Engineering Research Council of Canada and the Fonds de recherche du Québec – Nature et technologies. We also thank the Digital Research Alliance of Canada (alliancecan.ca) for the use of their computing resources.

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

## A  Implementation Details

### A.1  Datasets and Hyperparameters

For all datasets in our experiments, we split training data 50/50 to use as training/validation. Unless otherwise mentioned, our experiments were run 3 times to report average and standard deviations. To tune hyperparameters, we performed either grid search or used similar hyperparameters when comparing with other papers.

For all our experiments $\beta = 0.95$. To train supernet on pooling search space (Javan et al., 2023) for our experiments for classification task, we used SGD with learning rate 0.1 with cosine annealing, weight decay $1e-2$ and batch size 256 and we train for 500 epochs. For experiments on NAS-Bench-201 (Dong and Yang, 2020), we train for 50 epochs with SGD with learning rate 0.025 and cosine annealing. For image segmentation on Auto-DeepLab, we use same hyperparameters as original paper, using SGD with initial learning rate 0.025 decayed by annealing and weight decay 0.0003. Furthermore, we utilize mixed-precision operations and FFCV (Leclerc et al., 2023) library to accelerate training in our experiments.

For the benchmarks for classification, we directly reported the searched architecture performance. For segmentation task, we retained all architectures in table 4 with same setting as Liu et al. (2019).

### A.2  Boltzmann Softmax Exploration (BSE)

Boltzmann sampling is one of the simplest reinforcement learning exploration strategies. The probability of sampling architecture $a$ is defined as:

$$P(a) = Softmax(A(a)/T) \tag{4}$$

where $A(a)$ is the reward (validation accuracy) and $T$ is the temperature, controlling exploration/exploitation trade-off. In our experiments, we linearly decrease $T$ as $1 \rightarrow 1/100$.

