# OpenReview forum: "Iterative Monte Carlo Tree Search for Neural Architecture Search"
_automl.cc/AutoML/2025/Methods_Track — AutoML 2025 Methods Track_

### Official Review · Reviewer_efJM · 2025-04-29

**Comments To Authors:**

**Summary:**
This paper introduces a new algorithm that uses MCTS and an iterative method for finding the right node visiting order. This is applied to a one-shot NAS setting. The MCTS algorithm effectively reduces the size of the supernet that we consider. While previous work uses a manual node visiting order, this method learns a better order in between sampling/supernet training. The key novelty is that this method allows the tree structure to be learned along with the supernet.

**Claims:**
* *We show that by iterative application of MCTS and tree reorganization, we can obtain competitive architectures without the prohibitive cost or constraints of previous methods.* This claim is evidenced.
* *We show that with a careful balance of exploration and exploitation, the number of iterations needed is small and the overhead cost is negligible.* This claim is evidenced.
* *We empirically validate our method for two computer vision tasks of image classification and semantic segmentation on three datasets. We show that compared to other MCTS-NAS methods that perform supernet training and MCTS jointly, our approach achieves competitive performance with linear computational complexity.* This claim is evidenced.

The claims are overall well evidenced by the experimental results.

**Strengths:**
* I like the idea of iteratively updating the tree structure to encourage good or promising subnets to be sampled.
* The method can improve one-shot NAS by focusing the search towards promising parts of the search space.
* The method is evaluated in a few different settings, including understudied segmentation.

**Weaknesses:**
* There are many improved baselines this work could compare to, building on e.g. DARTS and zero-cost proxies.
* Is it not very expensive to obtain all accuracies for all leaves in the tree T? Would this not involve evaluating every possible architecture in the search space? How much time does it take to obtain these and is it done once in the beginning and then these accuracy estimates are dynamically updated by the MCTS? What is the cost associated with each method in Table 3 (right)?
* How does this algorithm scale to larger search spaces?
* It is a bit unclear to me what happens when the tree is reorganised. How are the existing n-visits and rewards reused in the new tree?
* A visualisation of the tree structure updating would help a lot in understanding Section 3.3.
* “…and the training of the supernet is started again with the new tree structure” Does this mean the supernet is reinitialised and trained from scratch, or that supernet training resumes, just with a new tree structure to sample subnets from? If the latter, then change it to “…training of the supernet resumes with the new tree structure”.
* I don’t understand the line “𝑎 ← 𝑖∗” in Algorithm 1. Should this not be “𝑖 ← 𝑎”?
* When competing the rank correlation of the top-k architectures in Pooling benchmark, k=10 is a very low number. To get more certainty on the scores k should be increased.
* The paper presents no insights on what nodes are selected as important by the tree reorganisation. This would help give insights.

**Minor:**
* Remove “for” on line 10
* Remove “has” on line 38
* Change “he” to “the” on line 95
* Fix broken citation, “spaceWang et al. (2020a)”, on line 202
* Fix “…and is require not further validation…” on lines 264-265
* Repetition on lines 274-275.

I think that this method needs to be compared with newer baselines, though I appreciate the main contribution is to build upon other MCTS approaches and that as far as I know, these MCTS baselines are fair. I overall lean slightly towards acceptance.

**Review Confidence:**

4

**Review Rating:**

7

---

### Official Review · Reviewer_YU6z · 2025-04-30

**Comments To Authors:**

Factual aspects:
- State/summarize the main contributions of the paper in a few sentences.
This paper introduces a MCTS for NAS framework that uses tree reorganization, Boltzmann sampling, and balanced exploration-exploitation within the search space. Each of these are varied improvements upon prior work using MCTS for NAS that are competitive with the state of the art.

- Compare the paper with previous work. In particular, is there highly relevant previously published work that the authors do not seem to be aware of?
The author referenced critical prior work in NAS space and specifically for MCTS for NAS.
- Express your level of confidence in the correctness of the results, and point out any major errors, if any are found.
I am 85% confident in the correctness of the results shown in this paper. While there are no glaring errors in the methodology, there are a number of grammatical mistakes that could do with a review. (e.g. p1-L38, p2-L56, p2-L65)

- Final assessment:
- What are the strengths of the paper? (results? new research direction? application? etc.)
The strengths of this paper revolve around its clear communication of the design of this algorithm, and how each piece works to improve on prior work within the research space. This paper also showcases results competitive with the state of the art in a joint efficiency and accuracy metric.

- What are the weaknesses of the paper?
The weaknesses of this paper include a small scope; this paper is limited to MCTS for NAS rather than a comprehensive comparison against supernet-sampling NAS algorithms. To follow on to this, the final results for this algorithm only have a modest improvement over previous papers within the space. Another weakness of the paper is that within the results section, the most recent paper is from 2023, with the rest from 2021 or prior. While it works to showcase a baseline for this algorithm, it does not allow for a good comparison against current methods.

- Express and explain your opinion regarding whether the contributions of the paper (assuming they are correct and original) are interesting/useful/relevant.
This paper introduces a number of improvements over prior MCTS for NAS algorithms. This novel algorithm is useful and releveant, however it is limited in scope and general application.

- Final Recommendation: Give a final recommendation for acceptance/rejection (or a more refined distinction, such as borderline).
Reject. This paper lacks a strong experimental comparison against similar techniques within NAS to showcase its potential usefulness. While it is definitely an improvement over prior techniques, it is not shown to be comparable to recent state of the art NAS techniques.

- Additional feedback: Comment on the quality, clarity, and readability of the writing. Provide comments that may help the authors in producing a revised version of the paper.
The paper did a wonderful job explaining the design of the algorithm and the reasoning behind it. While there were some issues with the readability at times, it was generally understandable and clearly presented the findings.

**Review Confidence:**

3

**Review Rating:**

3

---

### Official Review · Reviewer_g4DV · 2025-05-01

**Comments To Authors:**

## General Comments

This paper proposes an iterative Monte Carlo Tree Search (MCTS) approach for Neural Architecture Search (NAS). This paper proposes iterative approach that jointly trains with MCTS and learns the optimal node order in the search tree. It refines the tree structure at each iteration based on the estimated average accuracy of nodes on the validation set, aiming to improve sampling efficiency and guide the search towards high-performing architectures. Experiments on image classification and semantic segmentation tasks demonstrate the effectiveness of the proposed method.

## Pros

1. The concept of iteratively refining the tree structure based on the supernet's performance estimates is rational. Since the initial performance estimates may be inaccurate, this iterative process can gradually adjust the tree to prioritize promising architectures.

2. Using MCTS with UCT to sample architectures from the supernet and training the supernet on these sampled architectures is a well-founded approach. The use of Boltzmann sampling to balance exploration and exploitation further enhances the sampling process.

3. The experimental results show that the proposed iterative MCTS method outperforms other methods in terms of accuracy on both classification and segmentation tasks.

4. The ablation studies provide valuable insights into the behavior of the method.

## Cons

1. The method assumes a relatively good initialization of the supernet. In the experiments, using uniform sampling for initialization worked adequately, but this may not be the case in all scenarios. I want to see more discussions on this point.

2. Although the authors claim that the overhead cost is negligible, the iterative process still requires multiple rounds of supernet training and tree structure updates. The computational cost should be discussed in detail.

3. Recently, some articles have also used MCTS to solve black-box optimization problems (which can be also used to NAS), such as from the perspectives of high-dimensional optimization [1,2] and transfer optimization [3]. I would like to see more discussion on the differences between these works and this one, as including this would significantly enhance the quality of the paper.

[1] Learning search space partition for black-box optimization using Monte Carlo tree search. NeurIPS, 2020.

[2] Monte Carlo Tree Search based Variable Selection for High Dimensional Bayesian Optimization. NeurIPS, 2022.

[3] Monte Carlo Tree Search based Space Transfer for Black Box Optimization. NeurIPS, 2024.

**Review Confidence:**

5

**Review Rating:**

7

---

### Official Review · Reviewer_VKkD · 2025-05-01

**Comments To Authors:**

The paper proposes a new method for Monte Carlo Tree Search based NAS which better manages to adjust the search space to the results collected.


Questions:
- What is happening in figure 1 with the coloured symbols? Can you explain it? Should not red and orange go together as the highest scoring ones?
- Can you bring figure 1 into your explanation in section 3.3?
- Does the method presuppose a finite, enumerated search space?
- Line 190: Can you be more precise than the whole of section 4?
- A big part of the problem / solution seems to be the exploration/exploitation trade-off. Did you tune the hyperparameter C or did you set it to 0.5 arbitrarily? If you did, what was the impact?

Cons:
- Tables 1, 2 and 4 are inconsistent in the precision of the reported results. Sometimes there are two decimal digits, sometimes one. What does the tilde in table 2 mean? Can you update all the tables to be consistent in the number of digits?

Minor:
- Typos etc in lines 38, 65, 95, 192, 218, 219, 223, 255, 259
- Can't follow "is require not" in line 264, also estimate singular

**Review Confidence:**

2

**Review Rating:**

8

---

### Meta-Review · Area_Chair_kUSt · 2025-05-08

**Recommendation:** Accept
**Confidence:** 4

**Metareview:**

All reviewers acknowledge the soundness of the proposed method and its core contributions: integrating MCTS with supernet training in an iterative loop and dynamically refining the tree structure. The approach is well-motivated and effectively balances exploration and exploitation through Boltzmann sampling and UCT, which leads to competitive perfromance. Claims are well supported and the reproducibility review confirms completeness and clearness of the code.

The main concerns across the reviews revolve around the scope of the empirical evaluation. In particular there is a lack of comparisons to more recent and stronger NAS baselines. Also missing citations and comparisons in the related work are mentioned. Furthermore, there are questions raised regarding the computational cost and scalability of the method.

Despite these weaknesses, which I would expect the authors to address as best as they can in a final version of the paper, I recommend acceptance, given the novelty, strong empirical perfromance on established tasks, and the high reproducibility of the work.